# Specialist healthcare services for concussion/mild traumatic brain injury in England: a consensus statement using modified Delphi methodology

Elika Karvandi [iD],[1] Adel Helmy,[1] Angelos G Kolias [iD],[1] Antonio Belli,[2] Mario Ganau,[3] Clint Gomes,[4,5] Michael Grey,[6] Michael Griffiths,[7,8] Timothy Griffiths,[9,10] Philippa Griffiths,[11] Damian Holliman,[12] Peter Jenkins,[13,14] Ben Jones [iD],[15,16] Tim Lawrence,[3] Terence McLoughlin,[4] Catherine McMahon,[17] Shrouk Messahel [iD],[18] Joanne Newton,[19] Rupert Noad,[20] Vanessa Raymont [iD],[21] Kanchan Sharma,[22] Richard Sylvester,[23,24] Daniel Tadmor,[25,26] Peter Whitfield,[20] Mark Wilson,[14,27] Emma Woodberry,[28] Michael Parker,[29] Peter John Hutchinson [iD] [1]

For numbered affiliations see end of article.

**Correspondence to**
Elika Karvandi;
eek34@cam.ac.uk

## ABSTRACT

**Objective** To establish a consensus on the structure and process of healthcare services for patients with concussion in England to facilitate better healthcare quality and patient outcome.

**Design** This consensus study followed the modified Delphi methodology with five phases: participant identification, item development, two rounds of voting and a meeting to finalise the consensus statements. The predefined threshold for agreement was set at ≥70%.

**Setting** Specialist outpatient services.

**Participants** Members of the UK Head Injury Network were invited to participate. The network consists of clinical specialists in head injury practising in emergency medicine, neurology, neuropsychology, neurosurgery, paediatric medicine, rehabilitation medicine and sports and exercise medicine in England.

**Primary outcome measure** A consensus statement on the structure and process of specialist outpatient care for patients with concussion in England.

**Results** 55 items were voted on in the first round. 29 items were removed following the first voting round and 3 items were removed following the second voting round. Items were modified where appropriate. A final 18 statements reached consensus covering 3 main topics in specialist healthcare services for concussion; care pathway to structured follow-up, prognosis and measures of recovery, and provision of outpatient clinics.

**Conclusions** This work presents statements on how the healthcare services for patients with concussion in England could be redesigned to meet their health needs. Future work will seek to implement these into the clinical pathway.

## INTRODUCTION

In England and Wales alone, more than a million people suffer a head injury every year.[1] These injuries occur as a result of direct or indirect impacts to the head and

## STRENGTHS AND LIMITATIONS OF THIS STUDY

⇒ This consensus statement outlines structural and process-based factors needed to move towards standardised best-practice care for concussions in England to improve healthcare quality and patient outcomes.

⇒ To our knowledge, no studies have reported on the healthcare system for concussion in England.

⇒ We engaged with various stakeholders involved in concussion care to ensure different perspectives are represented.

⇒ It cannot be excluded that some disagreement on statements was in part due to differences in definitions and terminology use despite probable broad agreement on the statement in principle.

⇒ The statements reached agreement in this panel, but we cannot exclude the possibility or to what degree the results are influenced by the experience and expertise of these individuals both for included and excluded statements.

are commonly encountered in sports, both at professional and grassroots levels. Injuries to the head, particularly in the context of sport, are often labelled 'concussions'. While there is a recently published definition available,[2] there is no widespread consensus for what constitutes a concussion, and it is often used to describe both the symptoms suffered following a head injury as well as the pathophysiological mechanisms causing these symptoms.[3][4] Despite some controversies in the field, concussion is widely recognised by the public and those playing sport, and for this reason, we have used the term for the purpose of this paper. A concussion is diagnosed on clinical grounds with evidence of

immediate and transient neurological dysfunction that may or may not present with a wide range of signs and symptoms, such as loss of consciousness, post-traumatic amnesia, headache and vomiting.[5] However, most concussions do not present with obvious neurological signs, such as loss of consciousness, making recognition and diagnosis challenging.[6–9]

The post-concussion sequelae are assumed to recover spontaneously within a few days to a couple of weeks. Yet a proportion of individuals experience persistent symptoms for months to years after injury that requires timely and appropriate medical assessment and treatment.[10] Despite this known fact, guidance on clinical practice in the outpatient setting after the initial medial review is limited and ambiguous leaving the concussion population vulnerable to variations and suboptimal care.[11] This results in delayed intervention, resource waste, care inequalities and poorer quality of care.[12–14] There is no standardised, structured follow-up for those who experience persistent symptoms and current clinical services largely fail to meet the healthcare needs of these patients.[15]

Nonetheless, the past two decades have seen an exponential growth in concussion research which has improved our understanding of the injury and strategies to enhance recovery and quality of care.[16] Additionally, there is increasingly more awareness around the potential lasting effects of concussions, especially in sports, where there are growing concerns and calls for action.[17] The government and sports' national governing bodies are attempting to improve concussion management in all levels of sports by standardising protocols, allocating more funds for concussion research and incorporating new playing rules to reduce concussion risk and improve management.[18] Consequently, the demand for care is evolving and to meet patient needs changes to clinical services are required to provide evidence-based best-practice care while being cost-effective, delivering better healthcare value and quality and improving health outcome for patients with concussion.[7 19] This necessitates fundamental changes to the healthcare system in concussion care to facilitate the delivery of best-practice outpatient care and improve quality of care.[20] This consensus statement addresses the healthcare structure and processes in outpatient concussion care required to facilitate the delivery of best-practice care in specialist outpatient clinics to meet healthcare needs and improve patient outcome in the concussion population.

## METHODS

The guidance on Conducting and REporting DElphi Studies was followed for reporting this consensus study.[21]

The work was undertaken as part of wider work by the UK Head Injury Network. There was no direct involvement with patients or the public. The study did not meet the criteria for requiring ethical approval.

## Study design

A modified Delphi approach was used for this study. We followed a five-stage process: (1) panel identification, (2) literature search for initial statement development, (3, 4) two voting rounds followed by item removal and modifications and (5) consensus meeting and live voting to finalise the statements. The meeting was chaired by a working group member with previous experience in the Delphi consensus method.

## Panel selection

The UK Head Injury Network is a panel of clinical and academic experts with extensive experience in head injury care in England, and setup to improve access to and quality of concussion outpatient services—for sports-related and non-sports-related concussions—and provide a network to foster research collaborations. The network consists of a multidisciplinary panel from academia and clinical medicine with specialists in emergency medicine, neurology, neuropsychology, neurosurgery, neuroscience, paediatric medicine, rehabilitation medicine and sports and exercise medicine. Members are invited to join the UK Head Injury Network if they meet the following criteria based on peer-identification (all members in the network were identified or recommended by existing members): (1) healthcare professionals with extensive first-hand clinical experience in concussions care in any care setting or specialty; and/or (2) academic expertise in concussions. The group work with Love of the Game, a national sports concussion charity, to achieve these goals.

All members of the UK Head Injury Network at the time of the consensus process were invited to vote and attend the final consensus meeting. The participating members were geographically diverse including representatives from a range of healthcare settings across England. In addition to the various specialties involved, this ensures a more representative and informed decision-making for the different stages and settings of care across England.

## Evidence review

The need for a consensus statement had been discussed in previous meetings and highlighted in work completed by the group (unpublished observations). As the field of concussion has several contentious issues, we took a wide approach to topics covered in the initial statement development.

As the literature on implementation science in concussion is scarce, the literature search provided the evidence for clinical tasks that could be implemented. The panel used this literature combined with their first-hand local experience in various clinical settings to provide guidance on the structures and processes required to feasibly deliver these clinical tasks in England. Thus, statement development was guided by a literature search, topics of discussion within the network and results from previous work. Medline and Embase were searched for relevant literature in October 2022, primarily published national and international clinical guidelines and consensus

statements (see online supplemental material 1 for search strategy). The search was limited to literature published in English.

A subgroup of network members was responsible for this stage of the process (EK, PJH, AH). The initial statements were circulated to all network members for comments on the scope, structure and wording prior to starting the official rounds of voting.

## Consensus process

Two rounds of online surveys were administered between October and December 2022 followed by a virtual meeting in February 2023. The surveys were distributed for voting using the Qualtrics XM platform (Qualtrics, Provo, Utah).

All voting was anonymous to ensure uninhibited voting. Participants were asked to vote on each statement and had the binary options agree–disagree. The cut-off threshold for agreement was set at ≥70% for the two rounds of voting.[22–24] A statement was only included if agreement was above the predefined threshold for consensus. Free text comments were incorporated to allow members to suggest modifications or refinements of the statements. Statements that did not meet the threshold were excluded after each round of voting and statements were modified for further considerations if suggestions were made by the participants.

All participants were invited to a virtual meeting (Zoom Video Communications, San Jose, USA). The meeting was scheduled for 3 hours to provide sufficient time to discuss each statement in detail. The purpose of the meeting was to finalise the statements. Statements reaching agreement and disagreement were discussed where alternatives were considered for the latter kind. The threshold for agreement was raised to ≥85% for this meeting to ensure a robust degree of consensus. The final consensus statements were discussed and, where necessary, modifications were made to ensure clarity and accuracy before a final anonymous vote took place during the meeting.

## Patient and public involvement

No patient involvement.

## RESULTS
### Evidence review

While several papers were identified on therapeutic interventions, educational interventions and biomarkers for diagnosis and prognosis in concussion, as expected the literature on healthcare processes and structures in concussion is scarce. Thus, the final statements were primarily guided by previous discussions, interpretation of how research evidence on best-practice could be formulated into the structure and process of the healthcare system, and research work by the network and patient focus group sessions (unpublished observation).

Key papers identified during the literature review included clinical practical guidelines on concussion management[25–28] and systematic reviews on therapeutic interventions,[29–34] prognosis,[35–44] recovery[45 46] and biomarkers.[47 48] The clinical practice guidelines included a synthesis of practical guidelines primarily focused on four published papers: Center for Disease Control and Prevention Guideline on the Diagnosis and Management of Mild Traumatic Brain Injury Among Children (2019),[49] Consensus Statement on Concussion in Sport—the fifth International Conference on Concussion in Sport (2016), Ontario Neurotrauma Foundation Guideline for Concussion/Mild Traumatic Brain Injury & Persistent Symptoms (third edition; 2018), and Department of Veterans Affairs/Department of Defense Clinical Practice Guideline for the Management of Concussion-Mild Traumatic Brain Injury (2016).[26]

While it was agreed that covering a wide range of topics was appropriate during the initial statement development, the statements in this consensus report were narrowed down to specifically focus on the delivery of healthcare provisions through processes and structures in concussion care. This was deemed to be the most appropriate and relevant due to (1) some disagreement on clinical practice and management issues in concussion among the panel, (2) determining that these statements were beyond the purpose of this group and some of this guidance is featured in the most recent international *Consensus Statement on Concussion in Sport (sixth edition)*,[50] and (3) recognition that the need for the healthcare service to be improved at a system level prior to changes in day-to-day clinical practice. These points became clear after the first round of voting and the majority of these statement were removed after this stage. The decision to remove the remaining statements on day-to-day practice was made during the final meeting.

### Consensus process

Both rounds of voting had 21 votes (75% of network members at the time; see table 1 for demographic information on participating members). Following the initial statement development, the first round of voting had 55 items (figure 1, online supplemental material 2.1). Nine statements were eliminated after the first round of voting as they were below the threshold of agreement and six statements were amended based on comments from voting members. Of the 55 statements included in the first round of voting, 22 statements were included that did not follow the standard consensus methodology (online supplemental material 2.2). A different process was used for these statements and contained single-select or multi-select multiple choice options rather than the binary agree–disagree format used for the consensus statements. These statements are an important part of understanding the structure and processes for service delivery. Based on the responses from the first round of voting, three of these statements were amended and added as consensus statements with the binary agree–disagree response options in the following round of voting. As a result, the second round of voting included 26 statements that were

**Table 1** Demographics for participating panel members

| Characteristics | N (%) |
|---|---|
| Sex, female | 6 (21) |
| Specialty | |
| Emergency medicine | 2 (7) |
| Neurology | 3 (11) |
| Neuropsychiatry | 1 (4) |
| Neuropsychology | 3 (11) |
| Neurosurgery | 12 (43) |
| Paediatric medicine | 1 (4) |
| Rehabilitation medicine | 1 (4) |
| Neuroscience | 2 (7) |
| Sports and exercise medicine | 2 (7) |
| Other | 1 (4) |
| Region | |
| East Midlands | 1 (4) |
| East of England | 5 (18) |
| Greater London | 3 (11) |
| North East England | 4 (14) |
| North West England | 5 (18) |
| South East England | 4 (14) |
| South West England | 3 (11) |
| West Midlands | 1 (4) |
| Yorkshire and the Humber | 2 (7) |

put to a vote (online supplemental material 3). Three items did not reach the threshold of agreement following the second round of voting and were removed, leaving 23 statements to be discussed during the meeting. No statements were amended at this stage.

The virtual consensus meeting resulted in four statements being eliminated, two pairs of statements were combined and one statement was added. Of the eliminated statements, three were deemed to be beyond the scope of this consensus statement. The statement added during the meeting was suggested by one member and voted on twice as the level of agreement was just below the threshold following the first vote. The statement was modified and included following the second vote as the level of agreement was above the threshold.

This resulted in a final 18 statements (table 2), all above the raised threshold of 85% agreement. Fourteen statements achieved unanimous agreement. The final statements were organised into three themes: (1) care pathway to structured follow-up, (2) prognosis and measures of recovery, and (3) provision of outpatient clinics.

## DISCUSSION

Healthcare services for patients with concussion need to be transformed to advance quality of care and patient outcome. Current healthcare services fail to meet the needs of a significant number of patients with concussion which results in suboptimal recovery and failure of the healthcare system to fulfil its purpose of improving quality of life through enhancing health. There is an urgent

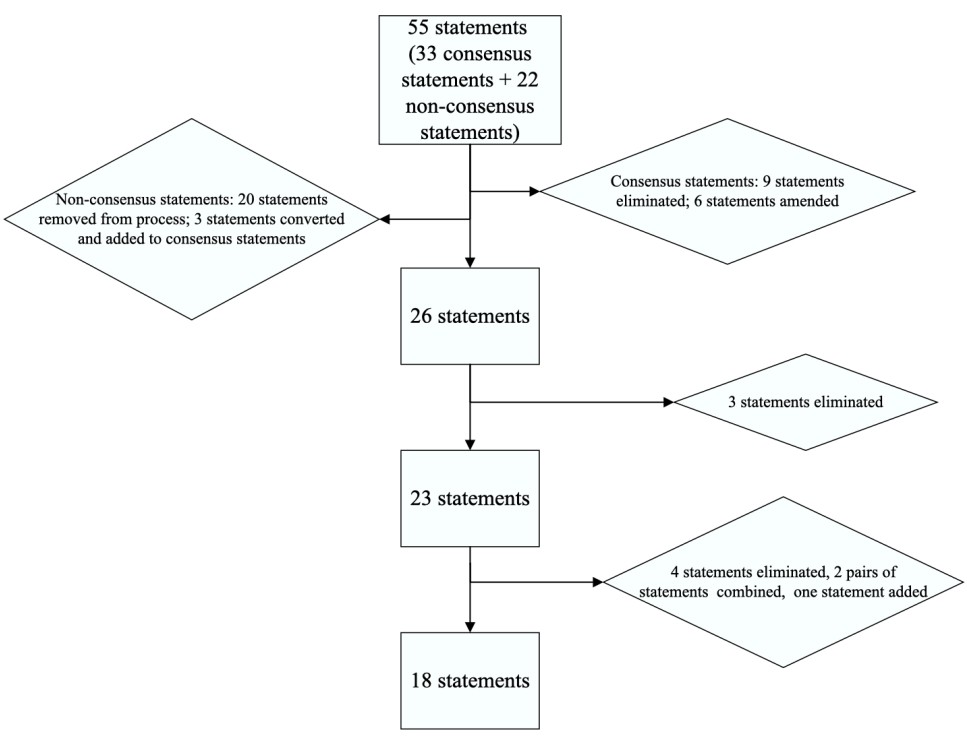

**Figure 1** Consensus process.

**Table 2** Final consensus statements

| | Agreement % |
|---|---|
| **1 Care pathway to structured follow-up** | |
| Patients with concussion with persistent symptoms should be able to access specialist clinics. | 100 |
| There should be dedicated concussion clinics, or where not possible, clinics with dedicated time for patients with concussion. | 100 |
| Specialist head injury clinics should comprise a multidisciplinary team that should include, as a minimum, a clinician, neuropsychologist and vestibular physiotherapist. | 100 |
| There should be a direct patient pathway for patients with persistent symptoms to specialist outpatient services from the ED, GP and other sources. | 100 |
| **2 Prognosis and measures of recovery** | |
| There should be a system for patients to be triaged to a specialist in head injury clinics based on persistent symptoms. | 100 |
| There may be a role for digital tools (eg, platforms, apps) to be used for screening for persistent symptoms following concussion. | 100 |
| Symptom scales can aid the assessment of symptom burden. | 100 |
| Screening for persistent symptoms can be aided by symptom scales. | 100 |
| A new patient-reported tool for the assessment of symptoms and outcome should be developed and validated. | 90 |
| Prolonged symptoms may develop even in the absence of specific acute signs, such as loss of consciousness, post-traumatic amnesia and vomiting. | 100 |
| There are no prognostic tools currently available with sufficient clinical accuracy to predict the risk of developing persistent symptoms. | 100 |
| There is currently insufficient evidence to support the clinical application of objective biomarkers (ie, fluid, imaging) for identifying and screening for persistent symptoms after a concussion. | 100 |
| Symptomatic patients with abnormal cranial/brain imaging should be considered for follow-up in a specialist clinic. | 89 |
| There is not yet sufficient evidence to support the routine use of advanced imaging techniques (eg, DTI, fMRI, MEG) in clinical practice. | 90 |
| **Provision of outpatient clinics** | |
| All patients should be directed to online information on concussion to help with expectations and recommendations. | 100 |
| All clinics should have access to standardised protocols for assessing pituitary dysfunction. | 100 |
| Comorbidities, socioeconomic status and social support should be factored into clinical decision-making on management. | 90 |
| The underlying cause(s) of persistent symptoms should be established and treated on a case-by-case basis. | 100 |

DTI, diffusion tensor imaging; ED, emergency department; fMRI, functional MRI; GP, general practice; MEG, magnetoencephalography.

need to improve the delivery of and access to concussion care in order to reduce healthcare inequalities and errors in management. Guidelines exist, but healthcare services need to restructure so to facilitate the delivery of these guidelines.

The recommendations in this consensus statement primarily focus on the structure and process of specialist healthcare services in concussion care with the purpose of improving healthcare quality and patient outcome.[20] These are the components in healthcare services that facilitate the implementation and delivery of best-practice care to achieve optimal health for the target patient population.[51] Ultimately, patient outcome is determined by the structure and process of care which describe the care context and the sum of actions in the delivery of care, respectively.[52] This consensus statement prioritises a patient-centric approach with a focus on timely, integrated, effective, efficient and evidence-based care and urge standardisation of these practices nationally.

Given the recent developments in the field, translating basic and clinical evidence into the healthcare system requires work on feasibility, implementation and service redesign.[53 54] Without a systematic approach, the promise and desired goal of improving patient outcomes can be challenging to fulfil. Simultaneously, there are still gaps in evidence. Consensus statements serve as an important tool in bridging areas where evidence is underdeveloped.[55] This consensus statement provides guidance outlines how to redesign the structures and processes in care delivery for patients with concussion in England.

This is through expert opinion based on the current level of evidence and first-hand local experience in the various clinical settings which patients with concussion encounter and academic expertise. The guidance needs to be updated as the evidence changes, but this ensures patient outcome to improve without delay.

The statements included in the first voting round that did not follow the consensus format will help guide system redesign in future work through providing further detailed guidance on fundamental factors in concussion healthcare delivery. This includes the structure of clinical teams, timelines for care tasks and other processes in the clinical pathway from injury to specialist outpatient services.

While the consensus statements are aimed at all concussions, sports-related and non-sports-related concussions, the implementation of these services into the National Health Services in England requires reallocation and reprioritisation of resources, as well as a restructuring of the system. Applying these changes for the subgroup of concussions occurring in sports—at both professional and grassroot levels—could be more practical and achievable at this stage, especially accounting for the high number of all-cause concussions. For example, direct patient pathways from local sports clubs could feasibly be implemented with quick-access clinics given the manageable number of sports-related concussions. A short-term goal of the UK Head Injury Network is the development of regional sports concussion clinics across the country to alleviate the burden of concussions in all levels of sports. These clinics aim to provide evidence-based care. However, it is intended that these services will ultimately be expanded and offered to all-cause patients with concussion. Thus, while the implementation of these statements will primarily change clinical services for sports-related concussions in the near future, the statements are directed at national health services offered to all patients in the longer term.

### Clinical implications
#### Care pathway to structured follow-up
Patients with persistent symptoms require further medical attention following the initial medical review—which is typically performed in emergency departments, general practice surgeries or by sport physicians—to recover to their preinjury level of health and function. Yet, in a Collaborative European NeuroTrauma Effectiveness Research in Traumatic Brain Injury (CENTER-TBI) study, 90% of 71 neurotrauma centres involved in the Europe-based study do not routinely schedule follow-up appointments for patients with head injury discharged from the emergency department.[56]

In the UK, there is no care pathway to ensure access to structured follow-up which is likely a fundamental underlying cause for the unmet needs in concussion care. The absence of such a service may be attributed to the heterogeneity of concussions and the vast differences in patient needs. However, a flexible and multidisciplinary approach is needed to personalise care and meet patient needs across the heterogenous recovery trajectories in the concussion population. Evidence suggests that providing multidisciplinary interventions can improve return to work and quality of life in this patient population.[57] Appropriate resources—time, equipment and staffing—need to be allocated to these services to ensure there is capacity to meet the diversity and scale of need. As a minimum, we suggest the inclusion of, though not limited to, a clinician (ie, specialist nurse, neurologist, neurosurgeon, sport and exercise physician), neuropsychologist and vestibular physiotherapist. This is due to the incidence of neurological, cognitive, psychological, cervicogenic and vestibular symptoms in post-concussive patients.[58–60] Moreover, due to the knowledge and skillset of physiotherapists, it may be beneficial to develop a new subspecialty in 'concussion physiotherapy'.[28 61]

Providing timely care is fundamental to meet patients' health needs. Delaying intervention can prolong recovery further and result in poorer outcome. One study found patients who sought care more than 30 days after a concussion had worse outcome at 3 months compared with patients who sought care earlier.[62] Concussion services should be redesigned with care pathways that offer access to structured follow-up for all patients with concussion as necessary to ensure these patients receive timely and appropriate care. These statements echo the recommendations in the recent Lancet Commission on traumatic brain injury.[63]

### Prognosis and measures of recovery
Objective measures of disease and prognostic factors provide a means for directing care.[64] One of the greatest challenges in concussion management is the lack of objective markers and the absence of reliable prognostic factors for recovery. Thus, early identification of patients at risk of prolonged recovery is not possible and therefore assessing the need for follow-up for individuals at an early stage is challenging. Consequently, no predictive tool can be used reliably in the clinical setting to assess the risk of prolonged recovery and the need for follow-up in the outpatient setting.[40] Even patients without marked acute signs and symptoms are at risk of prolonged recovery and should not be dismissed inadvertently. Of note, the panel recommends that patients with abnormal scans during the acute phase should always be offered follow-up as there is significant observable pathology during the acute phase to warrant further medical assessment.[65] There is evidence to suggest that some gross structural abnormalities found on acute CT imaging may be associated with worse outcomes.[66] However, we suggest that anything beyond standard structural imaging modalities—to rule out other pathologies—should not be used in the clinical setting. While advanced imaging methods are showing promise in research, they are yet to demonstrate reliable clinical utility and we do not recommend using these tools in clinical practice.[38 47 48]

In the absence of validated objective measures, ongoing self-reporting is currently the most reliable primary indicator of health status and method for identification of patients with persistent symptoms.[67] Of note, non-disclosure of concussion symptoms is a concern that needs to be addressed and is particularly prevalent in professional and community sports.[68 69] To filter out people who have recovered and identify individuals who need further medical review, we recommend a screening system. This would need to be cost-effective and scalable due to the size of the concussion population. We suggest digital tools as a platform for the screening system as they offer a remote, convenient, efficient and scalable solution.[70] Once set up, costs remain comparatively low while also overcoming traditional barriers related to geographical, financial and resource limitations.[71 72] Such a system could also be efficiently and effectively used to triage patients to the most appropriate clinician in the multidisciplinary clinic based on patient-reported symptoms and disabilities.

Standardising evaluation of recovery in the absence of objective markers is challenging. However, symptom scales may aid assessment of symptom burden and provide a means for screening patients, particularly as history taking and patient-reported symptoms are currently the only methods for measuring recovery.[67] To achieve this long-term, we recommend developing and validating a patient-reported outcome measure for the concussion population to help assess recovery and determine care needs. This may include assessment of symptom burden, functional status, return to work or activity and quality of life. However, the specifics need to be determined in an appropriate study. While this study focuses on clinically actionable statements, we believe this is a research priority that is important to include as it is required to achieve many of the clinical statements recommended. A validated patient-reported outcome measure will facilitate the clinical pathway for patients with concussion to specialist outpatient services and will help move the field forward.

In the era of patient-centric and value-based healthcare, patient-reported outcome measures are increasingly important to capture the patient's perspective of their health. Even when objective measures are identified in the future, these tools should be used to capture a holistic patient-centric assessment of health status and recovery.

### Provision of outpatient clinics

Providing early information and awareness is essential to managing expectations and can enhance recovery. A Transforming Research and Clinical Knowledge in Traumatic Brain Injury (TRACK-TBI) study found that only 42% of patients reported receiving head injury information at discharge from the emergency department.[73] Providing proper discharge information can be challenging in the emergency department due to the busy environment in these clinical settings and the cognitive state of the patients initially after injury which may prevent them from comprehending and retaining information.[74] Thus, we suggest directing patients to a single online platform for evidence-based information. The online modality allows anyone to access the information at any time, provides the same information to all regardless of geographical region and is easier to keep up to date with the evidence, which is particularly important in a fast-evolving field.

For patients with persistent symptoms, determining the most appropriate therapy can be challenging, especially in the absence of sufficiently strong evidence on therapeutic efficacy and timing of interventions. Given the vast heterogeneity in cases, no single treatment plan can be used for all patients with concussion and clinicians must evaluate the most appropriate intervention(s) on a case-by-case basis. While further research is required in this domain some therapeutic interventions have shown promise. Recommendations for therapeutic approaches are beyond the scope of this consensus statement and we recommend referring to other literature on this topic.[29–34] Biopsychosocial factors should be considered when assessing clinical and social care needs as these have been shown consistently to influence patient recovery.[41]

Lastly, we suggest that concussion clinics should have access to a standardised protocol for assessing pituitary function. Pituitary dysfunction can result in a range of different symptoms, many of which overlap with concussion symptoms, including fatigue, anxiety and mood disturbances. The literature on the incidence of pituitary dysfunction after concussions and screening protocols is still an area of active research. We suggest referring to a guidance paper by the British Neurotrauma Group on screening and management of pituitary dysfunction after traumatic brain injury.[75]

### Recent developments in sports concussion guidelines

The sixth edition of the consensus statement on concussion in sport was published this year.[50] This statement also advises that individuals with persistent symptoms after a concussion (suggested as symptom persisting for longer than 4 weeks) should undergo a multimodal evaluation, including the use of standardised, validated symptom scales for assessment and to guide referrals. The statement recommends referral to specialist clinicians ideally part of a multidisciplinary network of practitioners, where possible, for targeted treatment for persistent symptoms including *cervicogenic symptoms, migraine and headache, cognitive and psychological difficulties, balance disturbances, vestibular signs and oculomotor manifestations.* Regarding the multidisciplinary network, the statement proposes the inclusion of *sports medicine physicians, athletic trainers/therapists, physiotherapists, occupational therapists, sports chiropractors, neurologists, neurosurgeons, neuropsychologists, ophthalmologists, optometrists, physiatrists, psychologists and psychiatrists.* Furthermore, biopsychosocial and comorbidities should also be considered in the context of persistent symptoms after concussions. Additionally, similar guidance has been provided regarding the use of

advanced neuroimaging and other objective biomarkers in clinical practice.

## Limitations

The literature on concussion, though rapidly evolving, is contentious and there are still numerous areas of disagreement among the panel. As discussed above, a wide approach was taken for the initial statement development which was later narrowed to focus on the healthcare structure and processes, in part. It cannot be excluded that some disagreement on statements was in part due to differences in definitions and terminology use despite probable broad agreement on the statement in principle. However, establishing agreement on terminology is a different consensus process on its own and was beyond the scope of this study.

We engaged with different stakeholders including academics, clinicians from different specialties and charities, and drew from previous patient-focus group sessions (unpublished observations). To ensure more robust statements, we chose a high threshold for consensus. While these statements reached agreement in this panel, we cannot exclude the possibility or to what degree the results are influenced by the experience and expertise of these individuals both for the included and excluded statements. The number of participants was in line with recommendations for Delphi studies[76]; however, more representatives from each specialty involved could have benefitted the consensus process.[77] The study was also limited in female representation. However, due to the high level of agreement and consistency with the literature, we believe the statements can be integrated reliably in the national healthcare system.

General practitioners are typically the first point of contact for patients with persistent problems seeking medical attention after a concussion and are a key player in the patient pathway. While the perspective of general practitioners is crucial for the wider concussion healthcare services, this study focused on the structures and processes in specialist outpatient services. However, the involvement of general practitioners is crucial for implementation and redesigning the concussion pathway.

## CONCLUSION

This consensus statement outlines structural and process-based factors needed to move towards standardised best-practice care in concussion. This includes a final 18 statements on the care pathway for structured follow-up, prognosis and measures of recovery and provision of outpatient clinics. The statements are aimed at both the sports-related and all-cause concussion populations; however, they should be implemented for sports-related concussions as the first step before expanding to all patients with concussion given the greater feasibility of this approach. Further research and stakeholder engagement is needed to determine best-practice clinical concussion management in outpatient clinics and the

recommendations in this statement provide the structures and processes whereby best-practice patient-centred care can be delivered in a timely, integrated, efficient and effective manner.

**Author affiliations**
[1]Department of Neurosurgery, University of Cambridge, Cambridge, UK
[2]Inflammation and Ageing, University of Birmingham, Birmingham, UK
[3]Nuffield Department of Clinical Neurosciences, University of Oxford, Oxford, UK
[4]Royal Liverpool University Hospital, Liverpool, UK
[5]UK Sports Institute, Liverpool, UK
[6]School of Sport, Exercise and Health Sciences, Loughborough University, Loughborough, UK
[7]Department of Clinical Infection, Microbiology and Immunology, Institute of Infection, Veterinary and Ecological Sciences, University of Liverpool, Liverpool, UK
[8]Department of Paediatric Neurology, Alder-Hey Children's NHS Trust, Liverpool, UK
[9]Department of Cognitive Neurology, Newcastle University, Newcastle Upon Tyne, UK
[10]Institute of Neurology, University College London, London, UK
[11]Sunderland & South Tyneside Community Acquired Brain Injury Service, Northumberland Tyne and Wear NHS Foundation Trust, Newcastle upon Tyne, UK
[12]Department of Neurosurgery, Royal Victoria Infirmary, Newcastle upon Tyne, UK
[13]Wessex Neuroscience Centre, Southampton General Hospital, Southampton, UK
[14]Imperial College London, London, UK
[15]Carnegie Applied Rugby Research (CARR) Centre, Leeds Beckett University—Headingley Campus, Leeds, UK
[16]England Performance Unit, Rugby Football League Ltd, Leeds, UK
[17]Manchester Centre for Clinical Neurosciences (MCCN), Salford Royal Infirmary, Northern Care Alliance, Liverpool, UK
[18]Alder Hey Children's Hospital NHS Foundation Trust, Liverpool, UK
[19]Newcastle Upon Tyne Hospitals NHS Foundation Trust, Newcastle Upon Tyne, UK
[20]University Hospitals Plymouth NHS Trust, Plymouth, UK
[21]Department of Psychiatry, University of Oxford, Oxford, UK
[22]Department of Neurology, North Bristol NHS Trust, Westbury on Trym, UK
[23]National Hospital for Neurology and Neurosurgery, London, UK
[24]Institute of Exercise and Health, University College London, London, UK
[25]Carnegie School of Sport, Leeds Beckett University, Leeds, UK
[26]Medical, Leeds Rhinos Rugby League Club, Leeds, UK
[27]Department of Neurosurgery, Imperial College Healthcare NHS Trust, London, UK
[28]Department of Neuropsychology, University of Cambridge, Cambridge, UK
[29]Love of the Game, London, UK

**Acknowledgements** We would like to thank Love of the Game for their support with this work. Peter Hutchinson would also like to thank the NIHR Senior Investigator Award, Cambridge Biomedical Research Centre, Medtech Brain Injury Co-operative, Global Health Research Group on Acquired Brain and Spine Injury and Royal College of Surgeons of England. AH is supported by the Cambridge Biomedical Research Campus, the Royal College of Surgeons of England, and is a Theme Lead in the NIHR Brain Injury MedTech Collaborative.

**Contributors** All authors have made substantial contributions to the manuscript in the following ways: EK, AH, AK, MP and PJAH led the delivery and planning of this consensus statement. AB, CG, MGre, MGri, TG, PG, AH, DH, PJAH, PJ, BJ, AK, EK, TL, TM, CH, SM, JN, RN, VR, KS, RS, DT, PW, MW, EW were responsible for the conception and design of the study. AK moderated the consensus meeting. AB, CG, MGre, MGri, TG, PG, AH, DH, PJAH, PJ, BJ, AK, EK, TL, TM, CH, SM, JN, RN, VR, KS, RS, DT, PW, MW, EW participated in the acquisition, analysis and interpretation of data. EK drafted the manuscript and CG, MGa, TG, AH, PJAH, PJ, BJ, AK, EK, CH, RN, DT, PW, and EW revised it thoroughly. All authors gave final approval of the submitted manuscript version. EK, AH, and PJAH are the guarantors for the overall study.

**Funding** The authors have not declared a specific grant for this research from any funding agency in the public, commercial or not-for-profit sectors.

**Competing interests** AB serves as an expert advisor on concussion for England's Rugby Football Union, Football Association and Premier League. He also consults and has stocks in Marker Diagnostics LTD and licenses in Salivary Biomarker

of Concussion. AH served on the NICE Guideline Committee Trauma Guidelines for 2023. BJ consults for Premiership Rugby, London, and Rugby Football League, Manchester. DT consults for the Leeds Rhinos Rugby League Club and Hull City Tigers Football Club. MG serves on the advisory board for the Players Football Association. RN has undertaken work as a medicolegal expert in sports concussions. RS has served as an independent expert in legal cases in which head injury has occurred, some of these were mild head injuries. He also has received funding from the Rugby Football Association and Football Association for the Advanced Brainhealth project. TG has advised on legal cases where head injury has occurred, some of which were mild head injuries.

**Patient and public involvement** Patients and/or the public were not involved in the design, or conduct, or reporting, or dissemination plans of this research.

**Patient consent for publication** Not applicable.

**Ethics approval** The study was exempt from requiring ethical approval in accordance with the University of Cambridge Policy on the Ethics of Research Involving Human Participants and Personal Data.

**Provenance and peer review** Not commissioned; externally peer reviewed.

**Data availability statement** Data are available upon reasonable request.

**ORCID iDs**
Elika Karvandi http://orcid.org/0000-0003-1287-8200
Angelos G Kolias http://orcid.org/0000-0003-3992-0587
Ben Jones http://orcid.org/0000-0002-4274-6236
Shrouk Messahel http://orcid.org/0000-0003-0645-3070
Vanessa Raymont http://orcid.org/0000-0001-8238-4279
Peter John Hutchinson http://orcid.org/0000-0002-2796-1835

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
