## [Reviewer comments · BMJ Open]

ARTICLE DETAILS

TITLE (PROVISIONAL)	Specialist Healthcare Services for Concussion/Mild Traumatic Brain Injury in England: A Consensus Statement using Modified Delphi Methodology
AUTHORS	Karvandi, Elika; Helmy, Adel; Koliass, Angelos; belli, antonio; Ganau, Mario; Gomes, Clint; Grey, Michael; Griffiths, Michael; Griffiths, Timothy; Griffiths, Philippa; Holliman, Damian; Jenkins, Peter; Jones, Ben; Lawrence, Tim; McLoughlin, Terence; McMahan, Catherine; Messahel, Shrouk; Newton, Joanne; Noad, Rupert; Raymont, Vanessa; Sharma, Kanchan; Sylvester, Richard; Tadmor, Daniel; Whitfield, Peter; Wilson, Mark; Woodberry, Emma; Parker, Michael; Hutchinson, Peter

VERSION 1 – REVIEW

REVIEWER	Noah Silverberg The University of British Columbia
REVIEW RETURNED	14-Aug-2023

GENERAL COMMENTS	This Delphi study aimed to develop a consensus statement on “the structure and process of care for concussion patients in England.” The authors recruited a convenience sample of local experts and had them vote on statements. Statements were revised through two online Delphi rounds and then finalized at a live virtual meeting. This project has laudable goals, but the methodology is not well described, making it difficult to evaluate. Please see my specific concerns below. Delphi participants were invited from the UK Concussion Network. How does one become a member of the UK Concussion Network? How do authors know that Network members are representative of health professionals providing concussion care in the UK? Did authors apply an operational definition of “expert” to determine which members were eligible? Were all Network members invited, and of those, how many agreed? The authors say that “Published clinical practice guidelines were used to aid statement development” (pg 5). Which ones? How did the authors go about searching and selecting guidelines? They also stated that “systematic reviews and randomised controlled trials were used to determine whether the level of evidence was sufficient for implementation” (pg 5). Which ones? Did the authors use a systematic framework for appraising and synthesizing evidence? The authors reported that “Keywords used in the literature searches included concussion, mild...” (pg. 5). They should include their full
--

	search strategy in an appendix so readers who wish to can assess its rigor and thoroughness. “Key papers identified during the literature” (pg. 6). On what basis were these papers selected? How were agreement ratings obtained? Were participants given a binary choice – agree or disagree? The format and processes for the “virtual consensus meeting” are not sufficiently described. There are many apparent statements of fact in the Introduction and Discussion that need to be supported by cited evidence. For example: “current clinical services largely fail to meet the healthcare needs of these patients” (pg 3). As another example, the Prognosis and Measures of Recovery subsection of the Discussion runs multiple paragraphs without any citations. In other places, citations are provided, but more recent and reliable sources are available (e.g., “Yet a proportion of individuals experience persistent symptoms for months to years after injury...” on Pg. 3: see https://pubmed.ncbi.nlm.nih.gov/36472218/). There are references to work “for publication” (under review?) and “unpublished.” Please check BMJ standards for citing unpublished work. The consensus statements appear to include both clinically actionable recommendations and suggestions for research priorities (e.g., A new patient-reported tool for the assessment of symptoms and outcome should be developed and validated.) Why were statements of the latter type retained? Regarding “and some of this guidance would likely to be featured in the next international Consensus Statement on Concussion in Sport,” the 6th consensus statement was published in June 2023. This language could now be updated. The plain language summary would benefit from involvement of patient partners. It is unclear to what degree the various consensus statements are based on evidence vs. opinion. The authors performed a literature search for primary evidence, but there are insufficient details about how the results of that search were incorporated in the drafting or refinement of statements. “There is no consensus definition for what constitutes a concussion” (pg. 3). One was recently published: https://pubmed.ncbi.nlm.nih.gov/37211140/ Please check the formatting of the reference list.
--	---

REVIEWER	Bart Depreitere Universitaire Ziekenhuizen Leuven, Neurosurgery
REVIEW RETURNED	26-Aug-2023

GENERAL COMMENTS	The authors have conducted a Delphi consensus study to arrive at a set of statements on structure and process of specialist healthcare services for patients with persistent complaints following concussion.
---

	The statements should help to guide the required steps and reforms in healthcare organization to deliver better care to this vast group of patients. Following a methodologically sound process, the authors arrived at 18 statements. The authors should be commended for this work, which is timely and needed, not only in the UK, and could serve as template for actions also outside of the UK. I have only a couple of minor comments:  - In the introduction, the authors might consider spending a few words to explain the negative (direct and indirect) effects of suboptimal care for patients with persistent postconcussive complaints. - On p9 in the results section, the authors state that '12 (statements) were eliminated as to the format of the statements did not follow a consensus process'. This is, even after reading the supplementary material, insufficiently clear. - In the limitation section, the authors might comment on the low number of participants (21) and the absence of general practitioners and patients in the panel. - In the preparation phase, guidelines were used such as the Consensus Statement on Concussion in Sports 5. Meanwhile, a 6th edition of this Consensus Statement was recently published. The authors might spend a few words in the discussion on whether the new edition might have potentially led to different insights and statements.
--	---

VERSION 1 – AUTHOR RESPONSE

Reviewer: 1

Dr. Noah Silverberg, The University of British Columbia

Comments to the Author:

This Delphi study aimed to develop a consensus statement on “the structure and process of care for concussion patients in England.” The authors recruited a convenience sample of local experts and had them vote on statements. Statements were revised through two online Delphi rounds and then finalized at a live virtual meeting. This project has laudable goals, but the methodology is not well described, making it difficult to evaluate. Please see my specific concerns below.

Delphi participants were invited from the UK Concussion Network. How does one become a member of the UK Concussion Network? How do authors know that Network members are representative of health professionals providing concussion care in the UK? Did authors apply an operational definition of “expert” to determine which members were eligible? Were all Network members invited, and of those, how many agreed?

- Members are from a variety of specialties involved in concussion care ensures a representation of HCPs providing concussion care in England. We had at least one representative from every region in England and most regions had multiple representatives. Additionally, we had members from different settings (including communal care and rehabilitation centres, outpatient clinics in secondary care, tertiary hospitals, regional hospitals, academic and non-academic centres).
- Please note, while the network currently has representatives from all areas in the UK, the members at the time were only from England. The statements are applicable to the wider UK regions, however, due to the lack of representation from outside of England at the time the study was performed, we decided it to be more appropriate for the paper to only represent England.
- We have specified that network members were invited and the proportion of members who participated in the study.

The authors say that “Published clinical practice guidelines were used to aid statement development” (pg 5). Which ones? How did the authors go about searching and selecting guidelines?

- The search strategy has been added to the methods section and the guidelines used are referenced in the results section.

They also stated that “systematic reviews and randomised controlled trials were used to determine whether the level of evidence was sufficient for implementation” (pg 5). Which ones? Did the authors use a systematic framework for appraising and synthesizing evidence?

- These have been referenced in the results section

The authors reported that “Keywords used in the literature searches included concussion, mild...” (pg. 5). They should include their full search strategy in an appendix so readers who wish to can assess its rigor and thoroughness.

- We have now added the full search strategy in the supplementary material.

“Key papers identified during the literature” (pg. 6). On what basis were these papers selected?

- These papers were agreed by the core research team (EK, AH and PJA) for subsequent discussion. These papers directly addressed the question of specialised healthcare provision.

How were agreement ratings obtained? Were participants given a binary choice – agree or disagree?

- This has now been added and clarified in the methods section.

The format and processes for the “virtual consensus meeting” are not sufficiently described.

- We have further details on the virtual consensus meeting.

There are many apparent statements of fact in the Introduction and Discussion that need to be supported by cited evidence. For example: “current clinical services largely fail to meet the healthcare needs of these patients” (pg 3). As another example, the Prognosis and Measures of Recovery subsection of the Discussion runs multiple paragraphs without any citations. In other places, citations are provided, but more recent and reliable sources are available (e.g., “Yet a proportion of individuals experience persistent symptoms for months to years after injury...” on Pg. 3: see <https://pubmed.ncbi.nlm.nih.gov/36472218/>).

- We have added the appropriate references.

There are references to work “for publication” (under review?) and “unpublished.” Please check BMJ standards for citing unpublished work.

- We have now changed this to “unpublished observations” as per BMJ standards.

The consensus statements appear to include both clinically actionable recommendations and suggestions for research priorities (e.g., A new patient-reported tool for the assessment of symptoms and outcome should be developed and validated.) Why were statements of the latter type retained?

- We acknowledge that this is a research recommendation, however, the statement on validating a new PROM was included because it has a great impact on the process of care delivery for concussion patients. A PROM with appropriate and validated measures for the target population will facilitate the clinical service development recommended in this consensus statement, and is directly applicable to our aims in

this regard. We have added a comment about this in the discussion.

Regarding “and some of this guidance would likely to be featured in the next international Consensus Statement on Concussion in Sport,” the 6th consensus statement was published in June 2023. This language could now be updated.

- The manuscript has now been updated to include the 6th edition. We have also added a paragraph discussing the statement.

The plain language summary would benefit from involvement of patient partners.

- We have removed this statement and plan to do an organised patient and public engagement initiative for this and other work soon.

It is unclear to what degree the various consensus statements are based on evidence vs. opinion. The authors performed a literature search for primary evidence, but there are insufficient details about how the results of that search were incorporated in the drafting or refinement of statements.

- As this is a consensus statement, the statements are made through expert opinion based on the current available evidence and their experience in the local clinical setting. We have clarified this in the manuscript.

“There is no consensus definition for what constitutes a concussion” (pg. 3). One was recently published: <https://pubmed.ncbi.nlm.nih.gov/37211140/>

- Thank you for sharing this new publication with us. We have read and agree with the authors' definitions and hope this can clarify much of the uncertainty on both the clinical and research sides in the future. Unfortunately, there have been inconsistencies in the definition used which have contributed to much of the confusion in the field. Despite this important publication, this definition is not universally used and accepted. As this work was done before the publication of the cited manuscript, we feel it would be inappropriate to reference it as the definition used for this work. However, we have referenced the paper and will review the definition in our network and its use in future work.

Please check the formatting of the reference list.

- This has been amended.

Reviewer: 2

Prof. Bart Depreitere, Universitaire Ziekenhuizen Leuven

Comments to the Author:

The authors have conducted a Delphi consensus study to arrive at a set of statements on structure and process of specialist healthcare services for patients with persistent complaints following concussion. The statements should help to guide the required steps and reforms in healthcare organization to deliver better care to this vast group of patients. Following a methodologically sound process, the authors arrived at 18 statements. The authors should be commended for this work, which is timely and needed, not only in the UK, and could serve as template for actions also outside of the UK.

I have only a couple of minor comments:

In the introduction, the authors might consider spending a few words to explain the negative (direct and indirect) effects of suboptimal care for patients with persistent postconcussive complaints.

- This is a very important issue and we hope to draw more attention to this in our work. We have added sentence about this to give readers an overview of what these effects of sub-optimal care look like.

On p9 in the results section, the authors state that '12 (statements) were eliminated as to the format of the statements did not follow a consensus process'. This is, even after reading the supplementary material, insufficiently clear.

- This has now been clarified in the manuscript and the statements have been added separately in the supplementary materials to provide clarity.

In the limitation section, the authors might comment on the low number of participants (21) and the absence of general practitioners and patients in the panel.

- The role of general practitioners in concussion care is crucial and it is important that they are represented in the redesign of concussion services. We have added this to the limitations. We have also commented on the sample size in the limitations.

In the preparation phase, guidelines were used such as the Consensus Statement on Concussion in Sports 5. Meanwhile, a 6th edition of this Consensus Statement was recently published. The authors might spend a few words in the discussion on whether the new edition might have potentially led to different insights and statements.

- We have updated the manuscript to include the 6th edition and added a paragraph about overlapping statements between the two studies.

The authors hope that these revisions have significantly improved this manuscript.

VERSION 2 – REVIEW

REVIEWER	Noah Silverberg The University of British Columbia
REVIEW RETURNED	25-Oct-2023

GENERAL COMMENTS	The revised manuscript is much improved. The authors were responsive to reviewer comments. I particularly appreciate that the methods are described in more detail. I am still not clear how “extensive” clinical experience and academic “expertise” were defined. If by self-identification, the authors could just say so. Some phrases in the revised text have problematic grammar, such as: “While there is a recently published definitions available...”, “The meeting was schedules for 3 hours...”, “No patient involved.”, “The non-consensus statements...” The revised manuscript states: “Systematic reviews and randomised controlled trials were used to guide expert opinion to determine whether there was sufficient evidence...” but the literature search
--

	strategy was not designed to capture these kinds of studies. Unlike in formal clinical practice guideline development (e.g., with GRADE), it is also not clear how this evidence was evaluated, weighted, or considered in drafting recommendations. I recommend the sentence be removed. The authors should add as a study limitation that females were underrepresented on the consensus group.
--	---

VERSION 2 – AUTHOR RESPONSE

We would like to thank the reviewer for their thorough review of the manuscript. We believe their comments from both review rounds has improved the quality and transparency of the manuscript and we would like thanks the reviewers contributions to this process.

I am still not clear how “extensive” clinical experience and academic “expertise” were defined. If by self-identification, the authors could just say so.

- A further clarification has been added and we hope this will suffice in clarity.

Some phrases in the revised text have problematic grammar, such as: “While there is a recently published definitions available...”, “The meeting was schedules for 3 hours...”, “No patient involved.”, “The non-consensus statements...”

- These errors have been corrected

The revised manuscript states: “Systematic reviews and randomised controlled trials were used to guide expert opinion to determine whether there was sufficient evidence...” but the literature search strategy was not designed to capture these kinds of studies. Unlike in formal clinical practice guideline development (e.g., with GRADE), it is also not clear how this evidence was evaluated, weighted, or considered in drafting recommendations. I recommend the sentence be removed.

- This sentence has now been removed

The authors should add as a study limitation that females were underrepresented on the consensus group.

- This limitation has been added to the appropriate section.